# 3D Change Detection Using Adaptive Thresholds Based on Local Point Cloud Density

**Dan Liu** [1,2,*], **Dajun Li** [2], **Meizhen Wang** [3] and **Zhiming Wang** [2]

1    Key Laboratory for Digital Land and Resources of Jiangxi Province, East China University of Technology, Nanchang 330013, China
2    Faculty of Geomatics, East China University of Technology, Nanchang 330013, China; djli@ecut.edu.cn (D.L.); 201910705019@ecut.edu.cn (Z.W.)
3    Key Laboratory of Virtual Geographic Environment, Nanjing Normal University, Nanjing 210023, China; wangmeizhen@njnu.edu.cn
*    Correspondence: liudan@ecut.edu.cn; Tel.: +86-189-7098-4689

**Abstract:** In recent years, because of highly developed LiDAR (Light Detection and Ranging) technologies, there has been increasing demand for 3D change detection in urban monitoring, urban model updating, and disaster assessment. In order to improve the effectiveness of 3D change detection based on point clouds, an approach for 3D change detection using point-based comparison is presented in this paper. To avoid density variation in point clouds, adaptive thresholds are calculated through the *k*-neighboring average distance and the local point cloud density. A series of experiments for quantitative evaluation is performed. In the experiments, the influencing factors including threshold, registration error, and neighboring number of 3D change detection are discussed and analyzed. The results of the experiments demonstrate that the approach using adaptive thresholds based on local point cloud density are effective and suitable.

**Keywords:** 3D change detection; adaptive thresholds; point-based comparison; point clouds





## 1. Introduction

Change detection is a major topic of research in remote sensing, which plays an essential role in various tasks such as urban planning and environmental monitoring [1–4]. Three-dimensional change detection is a relatively new topic that extends change detection on 2D data to a 3D space. The demand of 3D urban modeling and updating are greatly increasing with the development and improvement of smart cities. Three-dimensional change detection can directly reflect 3D changes of objects, which is a significant desirability advantage for urban model updating [5–7], building and infrastructure monitoring and management [8–10], etc.

With the rapid development of three-dimensional laser scanning technology and the perfect progress of point cloud processing, point clouds acquired by airborne laser scanning and mobile laser scanning have been increasingly adopted in 3D change detection [11–17]. Moreover, point clouds can be generated through 2D data, such as UAV (Unmanned Aerial Vehicle) images [18–20], terrestrial images [21,22], and video sequences [23,24] in photogrammetry and computer vision. Therefore, change detection from point clouds is extremely useful in many fields. The approaches of 3D change detection based on multi-temporal point clouds can be divided to the following types, i.e., model-based comparison and point-based comparison.

(1) Model-based comparison: In order to detect changes from point clouds, this approach usually coverts point clouds to digital surface models (DSMs) and then determines changes by comparing DSMs. Murakami et al. [25] fulfilled change detection of buildings by subtracting one-point cloud-derived DSMs from another. Vogtle et al. [26] recognized building damages by comparing DSMs generated from multi-temporal airborne

laser scanning data. Similar technologies have been presented in other studies, such as Chaabouni-Chouayakh et al. [27], Stal et al. [12], and Yamzaki et al. [28]. However, this method cannot detect changes in an object's profile information since this technology mainly achieves change detection through height difference from DSMs. Some studies extracted changes by classifying DSMs derived from point clouds. Choi et al. [29] and Chaabouni-Chouayakh et al. [27] first detected change areas through the two DSMs generated from multi-temporal LiDAR data. The change areas were then classified as different objects, e.g., trees, vegetation, buildings, or grounds. Xiao et al. [30] first classified laser data-derived DSMs as different objects such as buildings and trees. The change detection was then completed based on the above classification. The accuracy of the technology depends on the object classification, which is a complex and time-consuming process.

(2) Point-based comparison: This approach directly calculates distances between points in LiDAR data. Memoli and Sapiro [31], Hyyppa et al. [5], and Antova [32] presented the point-to-point comparison framework. For each point in the detected point clouds, the distance to the nearest point in the referenced point clouds is computed. The changes are then estimated through a distance threshold. In order to improve the operating efficiency of the algorithm, Girardeau-Montaut et al. [33], Xu et al. [34], and other studies [35–37] performed direct comparisons with the octree structure. It is more practical for point cloud data because 3D changes can be obtained directly by point-based comparison. However, the approach is very sensitive to point cloud density. Additionally, the selection of threshold values is a key problem that influences the accuracy of change detection. Schutz and Hugli [38], and Hyyppa et al. [5] determined the changes from 3D data by a certain threshold, which was set based on empirical value. Nevertheless, some points may be skipped over or some errors may occur during distribution of the points in point clouds, which is generally nonuniform. Therefore, the certain threshold is not appropriate for change detection from the point clouds of density variation.

In this paper, adaptive thresholds based on local point cloud density are presented for point-based comparison. The *k*-neighboring average distance and local density in the point clouds are first calculated. The threshold values are then defined based on the *k*-neighboring average distance and the local point cloud density. The influencing factors including threshold, registration error, and neighboring number of 3D change detection are analyzed in the experiments.

## 2. Methods

In this paper, 3D change detection from point clouds is presented by point-based comparison. The *k*-neighboring average distance of each point in the detected point clouds and the local densities of the *k*-neighboring points are calculated. The distance thresholds for identifying changes are given, combined with the *k*-neighboring average distance and the local densities.

### 2.1. Preliminaries

Point-to-point comparison based on closest point distance is the simplest and fastest way to detect 3D change between two point clouds as it does not require gridding or meshing of the data or calculation of the surface normal [33]. The nearest neighboring distance is calculated as the distance between two points: for each point in the detected point clouds, the nearest point in the referenced point clouds is searched and their Euclidean distance is computed [39].

For two point clouds, PC1 and PC2 (PC1 is the detected point clouds, and PC2 is the referenced point clouds), at different periods, the nearest neighboring distance $d(p_{1i}, p_{2j})$ is computed between a point $p_{1i}(x_{1i}, y_{1i}, z_{1i})$ in PC1 and its nearest neighboring point $p_{2j}(x_{2j}, y_{2j}, z_{2j})$ in PC2.

$$d(p_{1i}, p_{2j}) = \|p_{1i} - p_{2j}\| = \sqrt{(x_{1i} - x_{2j})^2 + (y_{1i} - y_{2j})^2 + (z_{1i} - z_{2j})^2} \qquad (1)$$

The following expression can be used to determine changes between PC1 and PC2.

$$B(p_{1i}, p_{2j}) = \begin{cases} 1, & if \ |d(p_{1i}, p_{2j})| \geq T \\ 0, & if \ |d(p_{1i}, p_{2j})| < T \end{cases} \tag{2}$$

where $T$ is the threshold to determine whether changes occurred between PC1 and PC2. Generally, $T$ is set as a stationary value based on experience or the average distance of all the points in PC1 and PC2. However, the local density variation of the point clouds is not considered in these algorithms using a certain threshold or the average distance of all the points in the point clouds.

### 2.2. Adaptive Thresholds Based on Local Point Cloud Density

To take into account the local density variation of the point clouds, the threshold values are defined through the $k$-neighboring average distance and local point densities in the point clouds.

#### 2.2.1. K-Neighboring Average Distance

For a given point $p_i$ in the point clouds $P$, its $k$-neighboring points $p_{im}(m = 1, 2, \cdots, k)$ can be found (Figure 1).

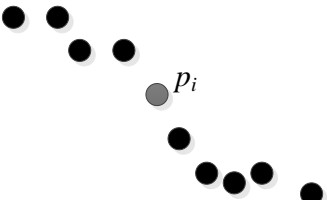

**Figure 1.** A point $p_i$ and its $k$-neighboring points $p_{im}(m = 1, 2, \cdots, k)$.

The local average distance $d_{ik}$ for the point $p_i$ and its $k$-neighboring points $p_{im}(m = 1, 2, \cdots, k)$ can be calculated as follows:

$$d_{ik} = \frac{\sum\limits_{m=1}^{k} \|p_{im} - p_{im}^N\|}{k} = \frac{\sum\limits_{m=1}^{k} \sqrt{(x_{im} - x_{im}^N)^2 + (y_{im} - y_{im}^N)^2 + (z_{im} - z_{im}^N)^2}}{k} \tag{3}$$

where $p_{im}^N$ is the nearest neighboring point of $p_{im}$ in the point clouds $P$.

Then, the $k$-neighboring average distance of all the points in $P$ is the mean value of the local average distances $d_{ik}$.

$$d_k = \frac{\sum\limits_{i=1}^{n} d_{ik}}{n} = \frac{\sum\limits_{i=1}^{n} \sum\limits_{m=1}^{k} \|p_{im} - p_{im}^N\|}{nk} \tag{4}$$

where $n$ is the number of points in $P$.

#### 2.2.2. Local Point Cloud Density

For the given point $p_i$ in the point clouds $P$, the local point cloud density index $I_{ik}$ can be estimated as follows:

$$I_{ik} = \frac{k}{\pi r_k^2} \tag{5}$$

where $k$ is the number of neighboring points to the point $p_i$ and $\pi r_k^2$ is the area of the circle centered at the point $p_i$ with a radius $r_k$ that is the fastest distance from the point $p_i$ to its $k$-nearest neighboring points (Figure 2).

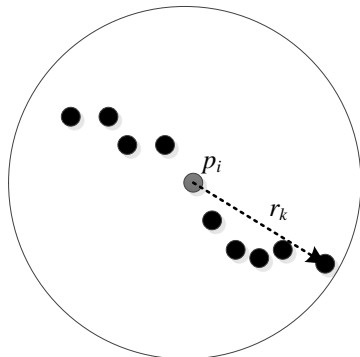

**Figure 2.** Calculation of the local point cloud density.

Then, the local point cloud densities $I_{ik}$ of all the points in $P$ are normalized by logarithmic function and rescaled to the range [0,1].

$$l_{ik} = \frac{\lg(I_{ik})}{\lg(\max(I_{ik}))} \quad (6)$$

According to the overall characteristics of the point cloud density, a larger value $l_{ik}$ means a higher density of the local points. Thus, the threshold for change detection should be less.

According to Equations (5) and (6), the thresholds can be expressed as follows:

$$T_{ik} = (\lambda - l_{ik})d_{ik} \quad (7)$$

where $T_{ik}$ represents the threshold of point $p_i$; $k$ is the number of neighboring points to point $p_i$; and $\lambda$ is a constant coefficient, which can be set to [1,3]. In the experiments presented in this paper, it is set as $\lambda = 2$.

### 2.3. Implementation of 3D Change Detection Using Adaptive Thresholds

For the compared point clouds PC1 and the referenced point clouds PC2, the overall algorithm of change detection for each point in the point clouds PC1 is as follows:

(1) Select a point $p_{1i}$ in the compared point clouds PC1.
(2) In the referenced point clouds PC2, search the nearest point $p_{2j}$ to the point $p_{1i}$.
(3) According to Equation (1), calculate the Euclidean distance $d(p_{1i}, p_{2j})$.
(4) In the point clouds PC1, search the $k$-neighboring points to the point $p_{1i}$.
(5) According to Equations (3) and (4), compute the $k$-neighboring average distance $d_k$ of PC1.
(6) According to Equations (5) and (6), compute the local point density $I_{ik}$ and its normalized value $l_{ik}$.
(7) Calculate the threshold $T_{ik}$ of point $p_{1i}$ using Equation (7).
(8) Detect whether point $p_{1i}$ has changed. If $d(p_{1i}, p_{2j}) \geq T_{ik}$, then a change in $p_{1i}$ has occurred.
(9) Use the above steps to detect changes in all the points in the point cloud PC1.

## 3. Experimental Results and Discussion

In order to determine the accuracy and effectiveness of the proposed approach, a series of experiments using test data was performed.

### 3.1. Experimental Data

In the experiments, the test data shown in Figure 3a were captured by the terrestrial laser scanner system Riegl-LMS-Z420i with a single shot accuracy of 10 mm at 50 m [40]. The test data were the building from the old gate of Tsinghua University in the city of

Beijing, China. To evaluate the performance of 3D change detection based on point clouds, the test data in Figure 3a were used as a temporal point cloud data named PC1. There were 168,603 points in the point clouds PC1. The other temporal data, PC2, shown in Figure 3b, were generated by deleting some points in PC1.

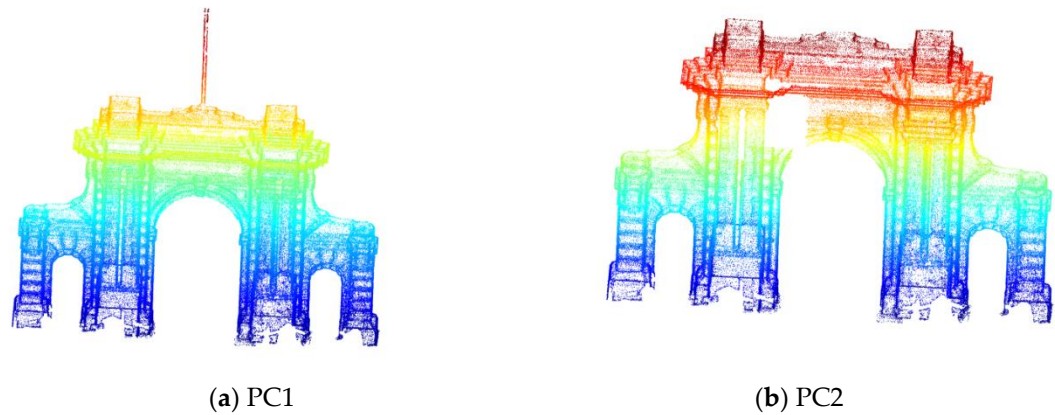

(**a**) PC1                 (**b**) PC2

**Figure 3.** The test data: (**a**) point clouds named PC1 and (**b**) PC2 generated by deleting some points from PC1.

Considering the influence of the threshold and neighboring number on the results of 3D change detection, several series of experiments with the test data were performed. To quantitatively assess the accuracies of the 3D change detection results on the test data, the following four quantitative measures were used: completeness, correctness, quality, and $F_1$ [41]. Completeness represents the proportion of correctly detected change points to the true change points. Correctness, which is the proportion of correctly detected change points to all the detected change points, estimates the reliability of 3D change detection. Quality and $F_1$ indicate the overall performance of change detection. They are defined as follows:

$$
\begin{aligned}
\text{completeness} &= TP/(TP+FN) \\
\text{correctness} &= TP/(TP+FP) \\
\text{quality} &= TP/(TP+FN+FP) \\
F_1 &= (2 * \text{completeness} * \text{correctness})/(\text{completeness} + \text{correctness})
\end{aligned}
\tag{8}
$$

where $TP$ (true position) is the number of correctly detected change points, $FP$ (false position) is the number of non-changed points incorrectly detected as changed, and $FN$ (false negative) is the number of change points falsely detecting non-changed points.

*3.2. Results and Discussion*

(1) Experiment 1 (Varying the registration error)

In the first series of experiments, 5% of the test data PC1 and PC2 were randomly sampled. After sampling, the average distance of PC1 was found to be 0.067 m. PC1 and PC2 were progressively misregistered by Gaussian noise with mean 0 and standard deviations 0, 0.004, 0.008, . . . , 0.076 *m* that was added to each point of PC1. It corresponds to the root mean square errors (RMSEs) $\sigma = 0$, 0.007, 0.014, 0.021, 0.028, 0.034, 0.041, 0.048, 0.055, 0.062, 0.069, 0.076, 0.083, 0.09, 0.098, 0.104, 0.111, 0.118, 0.124, and 0.133 *m* in total misregistration. Three-dimensional changes using adaptive thresholds in this paper were detected based on PC1 and PC2. The number of neighboring points was set as 50, i.e., $k = 50$. The results are shown in Figure 4 and Table 1. The qualitative results of 3D change detection with different registration errors are presented in Figure 5a–d.

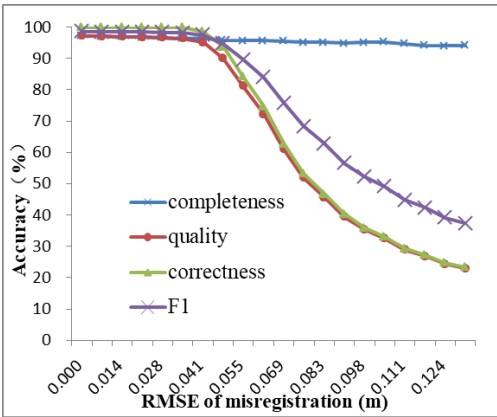

**Figure 4.** Results of experiment 1 (the results of 3D change detection varying the registration error through the approach using adaptive thresholds).

**Table 1.** The accuracy of 3D change detection with different registration errors through the approach using adaptive thresholds.

| Registration Error $\sigma(m)$ | Completeness (%) | Correctness (%) | Auality (%) | $F_1$ (%) |
|---|---|---|---|---|
| 0.048 | 95.78 | 93.71 | 90.01 | 94.74 |
| 0.055 | 95.74 | 84.19 | 81.15 | 89.60 |
| 0.062 | 95.74 | 74.64 | 72.22 | 83.88 |
| 0.069 | 95.58 | 62.73 | 60.96 | 75.75 |

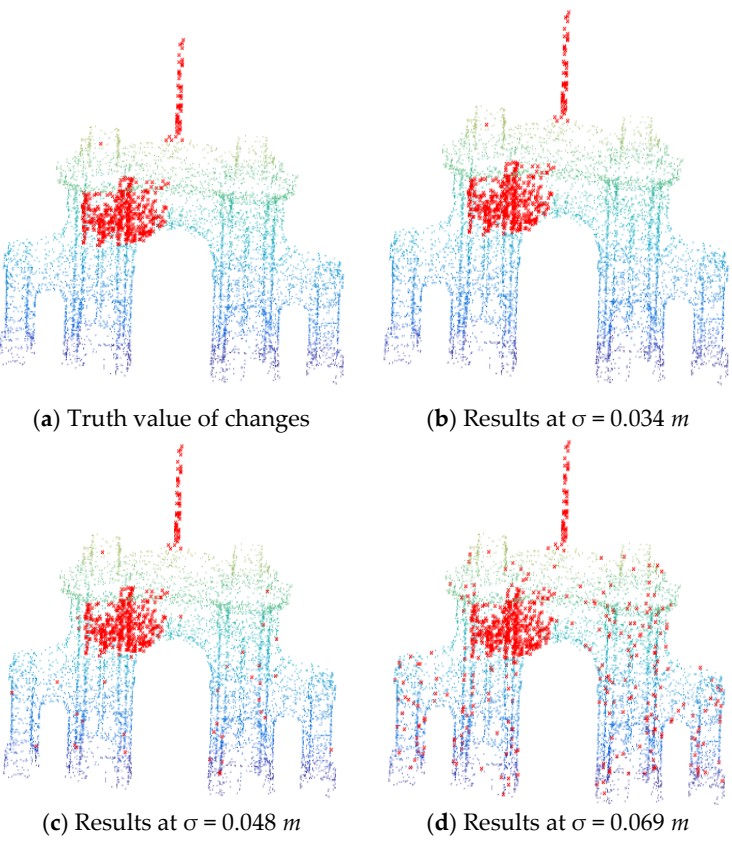

(**a**) Truth value of changes          (**b**) Results at $\sigma$ = 0.034 $m$

(**c**) Results at $\sigma$ = 0.048 $m$          (**d**) Results at $\sigma$ = 0.069 $m$

**Figure 5.** Qualitative comparison of 3D change detection at different misregistration of two point clouds (red points represent the change regions): (**a**) ground-truth of the change points and (**b**–**d**) results of change detection at the registration errors σ = 0.034 m, 0.048 m, and 0.069 m, respectively.

From the results illustrated in Figure 4, Figure 5, and Table 1, the following considerations can be outlined.

(a)　As the misregistration of two point clouds increases, the values of the indicators correctness, quality, and $F_1$ obtained through Equation (8) become lower. There is no evident changes in the value of completeness (completeness > 90% in all the errors of image registration), which means that most of the change points can be detected correctly. However, *FP*, i.e., non-changed points incorrectly detected as change points, becomes larger as the registration error increases. Therefore, misregistration of point clouds has significant effects on 3D change detection.

(b)　In Table 1 and Figure 5, when the registration error is $\sigma < 0.069\ m$, the method using adaptive thresholds can obtain stable and satisfactory results. That is to say, the registration error $\sigma$ should be less than the average distance of point clouds for 3D change detection using adaptive thresholds.

(2) Experiment 2 (Different threshold values *T*)

Like the first series of experiments, the processes of sampling and adding noise are done in this series experiments. To compare the results obtained using adaptive thresholds, the methods use the global average distance ($T = \dfrac{\sum\limits_{i=1}^{n} \|p_{1i} - p_{2j}\|}{n}$) and the local average distances (i.e., $d_{ik}$ calculated by Equation (3)). The results based on different threshold values *T* are shown in Figures 6a–d and 7a–d. In Figures 6 and 7, "global" represents the results obtained by using the global threshold, i.e., the average distance of all the points in PC1; "local" represents the results using the local average distances as the thresholds; and "adaptive" represents the results using adaptive thresholds in this paper. The number of neighboring points is set as 50 for "local" and "adaptive". The results presented in Figure 7b–d are obtained in the condition of misregistration $\sigma = 0.048\ m$.

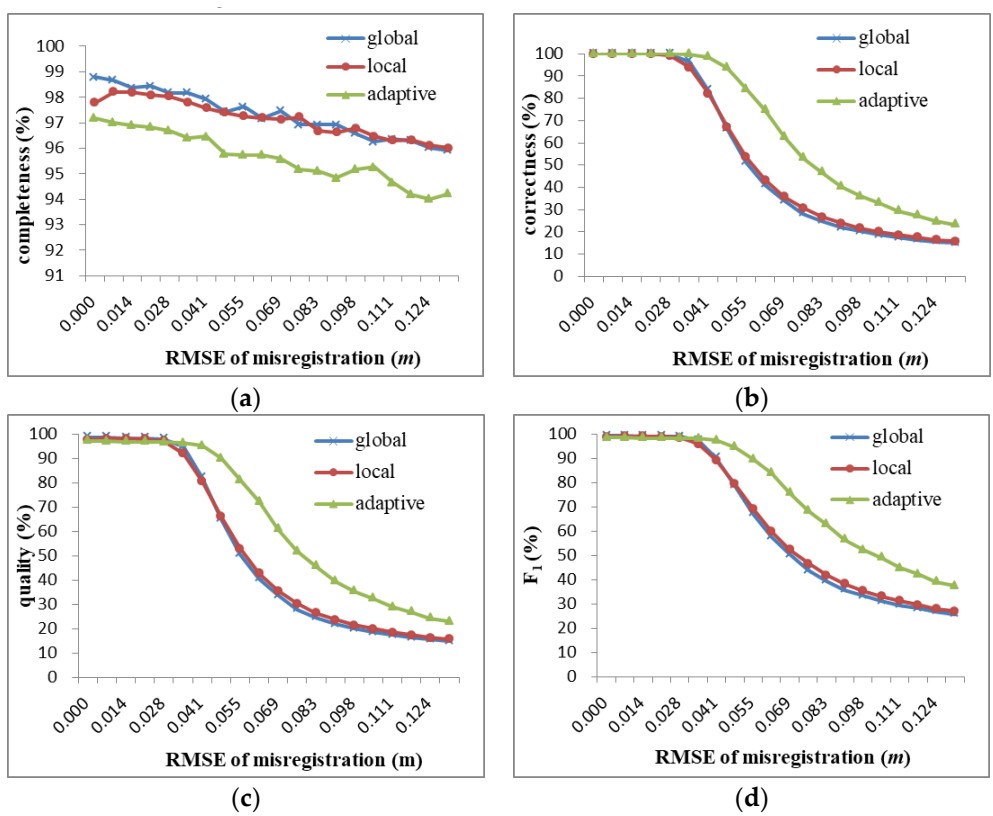

**Figure 6.** Results of experiments 2: (**a**–**d**) the results of the indicators completeness, correctness, quality, and $F_1$, respectively, through these methods.

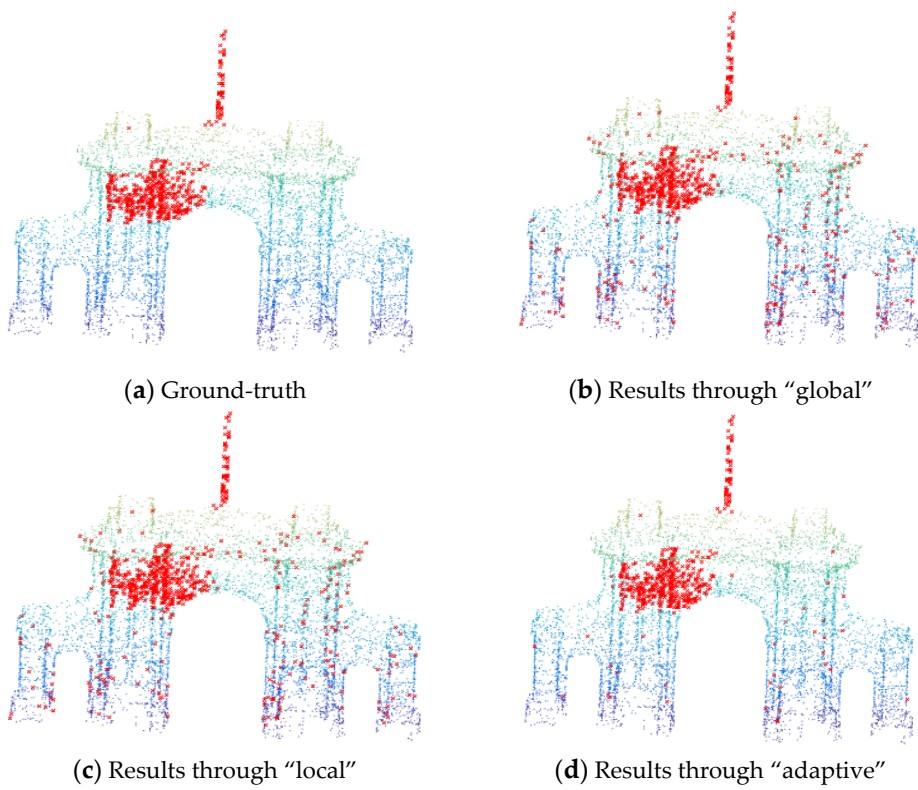

(**a**) Ground-truth

(**b**) Results through "global"

(**c**) Results through "local"

(**d**) Results through "adaptive"

**Figure 7.** Qualitative comparison of 3D change detection via the three approaches. Red points represent change regions. (**a**) Ground-truth of change points; (**b**) results of change detection through "global"; (**c**) results of change detection via "local"; and (**d**) results of change detection via "adaptive".

From the results illustrated in Figures 6 and 7, the following considerations can be remarked.

(a)  When the registration error $\sigma$ is very weak, all the methods using the above three thresholds can obtain appropriate accuracy. However, the method using the adaptive thresholds significantly outperforms the other methods with the increase in registration error $\sigma$.

(b)  When the registration error $\sigma$ is larger than 0.04 $m$, the performance of change detection using the global and local thresholds decline rapidly. Thus, the registration error $\sigma$ of the point clouds should be less than 1/2 of the average distance of point clouds for 3D change detection using the global and local thresholds. However, for the method using the adaptive thresholds, the registration error can be relaxed to the average distance of the point clouds. Therefore, the method using adaptive thresholds can obtain more satisfactory results than the two other methods.

(3) Experiment 3 (Varying the number of the neighboring points $k$)

In the third series of experiments, the test data were the same as that in experiment 1 except the number of neighboring points varied. The number of neighboring points was set as 10, 30, 50, 70 and 90. The results are presented in Figure 8a–d.

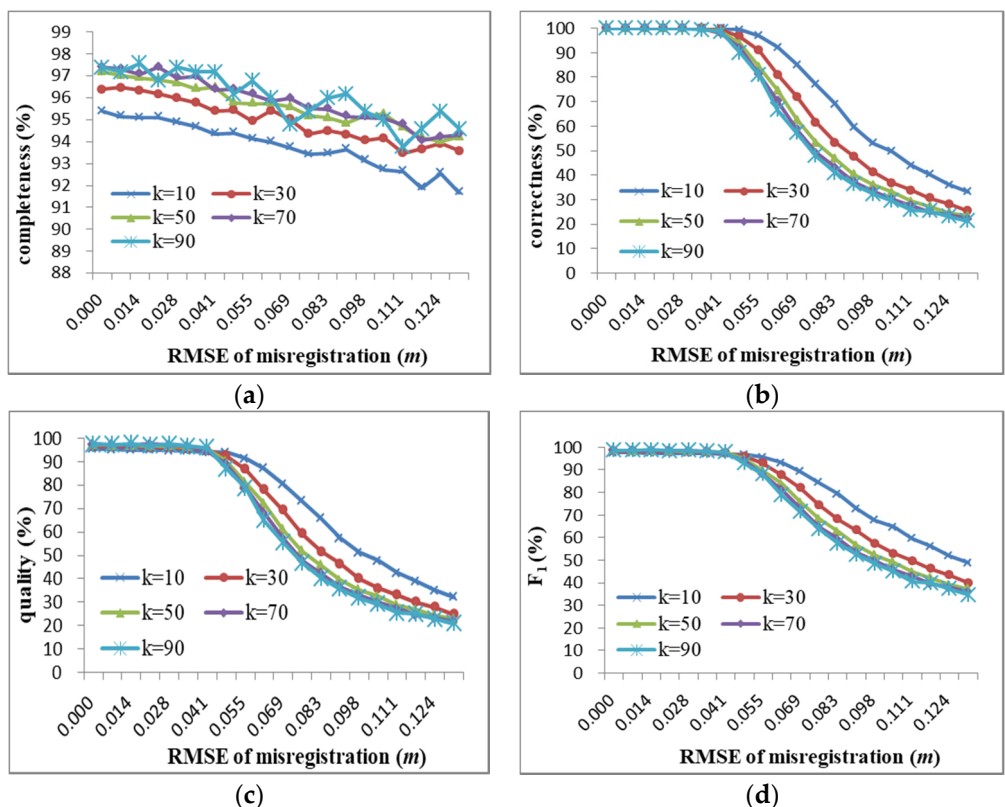

**Figure 8.** Results of experiments 3: (**a**–**d**) the results of the indicators completeness, correctness, quality and $F_1$, respectively, with varying number of neighboring points.

As the number of neighboring points increases, the method using adaptive thresholds has higher data accuracy, with the registration error being better than 0.041 *m*. However, the accuracy is lower with an increasing number of neighboring points *k* when the registration error is over 0.041 *m*. Moreover, the results of 3D change detection have no evident changes at $k > 50$. Therefore, the setting of *k* can be adjusted according to the registration error. *k* can be set to smaller values when the registration error is less than 1/2 of the average distance of point clouds. Instead, *k* can be set within a larger value (such as $k = 50$) when the registration error is larger than 1/2 of the average distance of point clouds.

## 4. Conclusions

In this paper, the development and implementation of 3D change detection based on a point-based comparison from point cloud data is presented. A particular feature of this approach is that adaptive thresholds are used to detect changes in the point clouds. To consider local density variation of the point clouds, the adaptive thresholds are calculated in combination with the *k*-neighboring average distance and the local point cloud density. Additionally, the influence of the registration error and the number of neighboring points on the accuracy of 3D change detection are investigated. A series of experiments on test data are presented in this paper. Compared with common methods with thresholds of global average distance and local average distance of point clouds, the experiments demonstrate that the approach based on adaptive thresholds is less affected by the registration error between point clouds. The experimental results show that the registration error for the approach using adaptive thresholds could be controlled within the average distance of the point clouds. Moreover, the number of neighboring points could select an appropriate value according to the registration error. Future work will focus on optimization of the algorithms in terms of the computation cost, etc.

**Author Contributions:** Dan Liu and Dajun Li conceived and designed the experiments; Dan Liu performed the experiments; Dan Liu and Meizhen Wang contributed to the analysis and interpretation of the results; Dan Liu wrote the manuscript; Dajun Li and Zhiming Wang revised the manuscript. All authors have read and agreed to the published version of the manuscript.

**Funding:** The work described in this paper was support by the National Natural Science Foundation of China (Project No: 41701437); by the Science and Technology Program of the Education Department of Jiangxi Province of China (Project No: GJJ180420); and by the Key Laboratory for Digital Land and Resources of Jiangxi Province, East China University of Technology (Project No: DLLJ201805).

**Institutional Review Board Statement:** Not applicable.

**Informed Consent Statement:** Not applicable.

**Data Availability Statement:** Publicly available datasets were analyzed in this study. This data can be found here: [http://vision.ia.ac.cn/data (accessed on 1 March 2021)].

**Acknowledgments:** Thank you to the China and National Laboratory of Pattern Recognition, Institute of Automation, Chinese Academy of Sciences, China. We also would like to express our gratitude to the anonymous reviewers for their comments and suggestions to this article.

**Conflicts of Interest:** The authors declare no conflict of interest.

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
