# Peer review of "3D Change Detection Using Adaptive Thresholds Based on Local Point Cloud Density"

_ijgi, doi:10.3390/ijgi10030127_

Round 1

Reviewer 1 Report

The paper is interesting, but the discussion section needs to be extended. The conclusions needs to be clarified. The literature revie needs to be updated.

The conclusion section should be extended. Please clarify the contributions of the work, practical and theoretical implications. The value of findings.

Most of references in the introduction are outdated. Please refer to new publications in this field. For example: 2019. Comparative analysis of machine learning and point-based algorithms for detecting 3D changes in buildings over time using bi-temporal lidar data. Automation in Construction105, p.102841. https://doi.org/10.1016/j.autcon.2019.102841

Line 263: how much less? Use values or percentages in this section to report the differences, or improvements in the current practice.

Section 3.2: discuss validation. How the results are validated and reliable?

The paper needs to be carefully edited to ensure there is no typos or errors. In line 220, there should be no (e) since there is no (e) in the figure.

The captions should be linger and informative. E,g, Figure 4 (line 191) the caption has no information “Results of experiment 1”. Best to mention what results, what data, what technique etc. in the caption.

Author Response

Point 1: The conclusion section should be extended. Please clarify the contributions of the work, practical and theoretical implications. The value of findings.

Response 1: A particular feature of the work is that adaptive thresholds are used to detect changes of the point clouds.  A further feature of the work is that the effects of the registration error and the number of neighboring points on the accuracy of 3D change detection are investigated by a series of the experiments.

To address this, the Conclusions section has been revised as:

“In this paper, the development and implementation of 3D change detection based on point-based comparison from Lidar point cloud data is presented. A particular feature of the approach is that adaptive thresholds are used to detect changes of the point clouds.  To consider local density variation of the point clouds, the adaptive thresholds are calculated by combining with the k-neighborhood average distance and the local point cloud density. Additionally,the influence of the registration error and the number of neighboring points on the accuracy of 3D change detection are investigated. A series of experiments on test data are presented in this paper. Compared with the common methods with thresholds of global average distance and local average distance of point clouds, the experiments demonstrate that the approach based on adaptive thresholds is less affected by the registration error between point clouds. The experimental results show that the registration error for the approach using adaptive thresholds could be controlled within the average distance of the point clouds. Besides, the number of neighboring points could select an appropriate value according to the registration error. Future work will focus on the optimization of the algorithms in terms of the computation cost, etc.

Point 2: Most of references in the introduction are outdated. Please refer to new publications in this field. For example: 2019. Comparative analysis of machine learning and point-based algorithms for detecting 3D changes in buildings over time using bi-temporal lidar data. Automation in Construction, 105, p.102841. https://doi.org/10.1016/j.autcon.2019.102841

Response 2: Corrected. The following publications in the field of change detection are referred in the Introduction section and added in the References section.

  1. Lin Y.; Zhang L.; Wang N.; et al. A change detection method using spatial-temporal-spectral information from Landsat images. International Journal of Remote Sensing, 2019, 41(2):1-22.
  2. Bolorinos J.; Ajami N. K.; Rajagopal R. Consumption Change Detection for Urban Planning: Monitoring and Segmenting Water Customers During Drought. Water Resources Research, 2020, 56(3).
  3. Matikainen L.; Pandi M.; Li F.; et al. Toward utilizing multitemporal multispectral airborne laser scanning, Sentinel-2, and mobile laser scanning in map updating. Journal of Applied Remote Sensing, 2019, 13(4).
  4. Wenzhuo L.; Kaimin S.; Deren L.; et al. A New Approach to Performing Bundle Adjustment for Time Series UAV Images 3D Building Change Detection. Remote Sensing, 2017, 9(6).
  5. Dong, P.L.; Zhong, R.F.; Yigit, A. Automated parcel-based building change detection using multitemporal airborne Lidar data. Surveying and Land Information Science, 2018, 77(1):5-13.
  6. Shirowzhan S.; Samad S.M.E.; Li H.; et al. Comparative analysis of machine learning and point-based algorithms for detecting 3D changes in buildings over time using bi-temporal lidar data. Automation in Construction, 2019, 105:102841.
  7. Santos, R.C.D.; Galo, M.; Carrilho, A.C.; et al. Automatic building change detection using multi-temporal airbone Lidar data. Proceedings of The International Archives of the Photogrammetry, Remote Sensing and Spatial Information Sciences, Volume XLII-3/W12-2020, 2020 IEEE Latin American GRSS & ISPRS Remote Sensing Conference, Santiago, Chile. 2020.

Point 3: Line 263: how much less? Use values or percentages in this section to report the differences, or improvements in the current practice.

Response 3: Thank you for your comments and suggestions. The results of the Experiment 2 in the section 3.2 show that the method using adaptive thresholds can obtain satisfactory results when the registration error less than the average distance of the point clouds, i.e. 0.067 m(F1=83.88% when σ=0.062 m; F1=75.75 % when  σ=0.069 m.). But the performance of change detection using the global and local thresholds are more affected by the registration error (for the method using local thresholds, F1=57.66% when  σ=0.062; F1=50.33 % when σ=0.069 m.).

Point 4: Section 3.2: discuss validation. How the results are validated and reliable?

Response 4: Thank you for your comments and suggestions. In the experiments of this paper, the test data used were from China and National Laboratory of Pattern Recognition, Institute of Automation, Chinese Academy of Sciences and published on http://vision.ia.ac.cn/data. To obtain the true changes, the test data named PC1 are used as a temporal point cloud data, and the other temporal data named PC2 are generated by deleting some points in the test data. Thus, the true change points can be gotten by subtracting PC2 from PC1. Additionally, to quantitatively assess accuracies of 3D change detection results on the test data, the following four quantitative measures are used: completeness, correctness, quality and . Therefore, the results in the paper are reliable.

To show the results more intuitively, the following Figure 5 and 7 are added in the experiment 1 and 2 respectively. Please see the attachment.

Figure 7. Qualitative comparison of 3D change detection via the three approaches respectively. Red points represent change regions. (a) Ground-truth of change points; (b) results of change detection through ‘global’; (c) results of change detection via ‘local’; (d) results of change detection via ‘adaptive’.

Point 5: The paper needs to be carefully edited to ensure there is no typos or errors. In line 220, there should be no (e) since there is no (e) in the figure.

Response 5: Corrected. Some other errors in the paper are found and revised.

Point 6: The captions should be linger and informative. E,g, Figure 4 (line 191) the caption has no information “Results of experiment 1”. Best to mention what results, what data, what technique etc. in the caption.

Response 6: Corrected. For example, the caption of Figure 4 is revised as “Results of experiment 1 (the results of 3D change detection with varying the registration error through the approach using adaptive threshold)”.

Reviewer 2 Report

It is an article that can be useful to those who expand their research
in this area.

However, the location of the test data must be mentioned,
because this fact offers quality and trust to the article.

At the same time, it would be good to present some details
about the analyzed objective, possibly images,
so that the reader has the image of the subject.

It is worth mentioning the need to rectify the inconsistent
use of the acronym "Lidar" vs "LiDAR", the second option being the most
appropriate.

The conclusions need to be expanded and improved
because the ideas are not developed enough.

Author Response

Point 1: the location of the test data must be mentioned, because this fact offers quality and trust to the article.

Response 1: The location of the test data is the Tsinghua University in the city of Beijing, China. To address this, the following description has been added in the first paragraph in the section 3.1.

“The test data is the building from the old gate of Tsinghua University in the city of Beijing, China.

Point 2: it would be good to present some details about the analyzed objective, possibly images, so that the reader has the image of the subject.

Response 2: Corrected.  To show the results more intuitively, the following Figure 5 and 7 are added in the experiment 1 and 2 respectively.Please see the attachment.

Point 3: It is worth mentioning the need to rectify the inconsistent use of the acronym "Lidar" vs "LiDAR", the second option being the most appropriate.

Response 3: Corrected.

Point 4: The conclusions need to be expanded and improved because the ideas are not developed enough. 

Response 4: A particular feature of the work is that adaptive thresholds are used to detect changes of the point clouds.  A further feature of the work is that the effects of the registration error and the number of neighboring points on the accuracy of 3D change detection are investigated by a series of the experiments. To address this, the Conclusions section has been revised as:

“In this paper, the development and implementation of 3D change detection based on point-based comparison from Lidar point cloud data is presented. A particular feature of the approach is that adaptive thresholds are used to detect changes of the point clouds.  To consider local density variation of the point clouds, the adaptive thresholds are calculated by combining with the k-neighborhood average distance and the local point cloud density. Additionally,the influence of the registration error and the number of neighboring points on the accuracy of 3D change detection are investigated. A series of experiments on test data are presented in this paper. Compared with the common methods with thresholds of global average distance and local average distance of point clouds, the experiments demonstrate that the approach based on adaptive thresholds is less affected by the registration error between point clouds. The experimental results show that the registration error for the approach using adaptive thresholds could be controlled within the average distance of the point clouds. Besides, the number of neighboring points could select an appropriate value according to the registration error. Future work will focus on the optimization of the algorithms in terms of the computation cost, etc.

Reviewer 3 Report

The presented work briefly illustrates the implementation of a particular method that allows to perform a change detection analysis on 3D point clouds. The research was conducted accurately, the methodology was described and the results were displayed appropriately. The experiments are described rigorously. Nevertheless, English style and correctness should be checked and revised in some portions of the paper.

In the following part there are some suggestions in order to improve the quality of the paper.

It may be appropriate to include some references in the introduction to photogrammetry and Structure from Motion techniques that allow you to generate 3D point clouds to which the same 3D change detection methods can be applied.

Line 157: the accuracy of the laser scanner reported is incorrect: from the cited reference it is described as “10mm@50m”

Line 175: “completebess” > “completeness”

Figures 5 and 6: there is a reference to (a)-(e) but only sub-figures up to (d) are displayed.

The References appear slightly dated. Apart from #18 from 2019, every other reference dates back to 2016. Considering that the subject is constantly evolving, some more recent innovations could be mentioned in the introduction.

In which development environment did you implement your method? Would it be interesting for you to release it to the scientific community on public domain platforms like GitHub?

Author Response

Point 1: It may be appropriate to include some references in the introduction to photogrammetry and Structure from Motion techniques that allow you to generate 3D point clouds to which the same 3D change detection methods can be applied.

Response 1: Corrected. The following description has been added in the second paragraph in the Introduction section.

“Besides, point cloud can be generated through 2D data, such as UAV images [18-20], terrestrial images [21, 22], video sequences [23, 24] in Photogrammetry and Computer Vision. Therefore, change detection from point clouds is extremely useful in many fields.”

  1. Eltner A.; Schneider D. Analysis of Different Methods for 3D Reconstruction of Natural Surfaces from Parallel-Axes UAV Images. The Photogrammetric Record, 2015, 30(151):279-299.
  2. Qu Y.; Huang J.; Zhang X. Rapid 3D Reconstruction for Image Sequence Acquired from UAV Camera. Sensors, 2018, 18(2):225-.
  3. Shahbazi M.; Menard P.; Sohn G.; et al. Unmanned aerial image dataset: Ready for 3D reconstruction. Data in Brief, 2019, 25:103962.
  4. Adilson; Tommaselli A.M.G. Automatic Orientation of Multi-Scale Terrestrial Images for 3D Reconstruction. Remote Sensing, 2014.
  5. Liu D.; LiuJ.; Wu Y.G. Depth Reconstruction from Single Images Using a Convolutional Neural Network and a Condition Random Field Model. Sensors, 2018, 18(5): 1318.
  6. Kundu A.; , Li Y.; Dellaert F.; et al. Joint Semantic Segmentation and 3D Reconstruction from Monocular Video. Proceedings of European Conference on Computer Vision, Zurich, 2014; Switzerland Springer International Publishing.
  7. Gerdes K.; Pedro M.Z.; Schwarz-Schampera U.; et al. Detailed Mapping of Hydrothermal Vent Fauna: A 3D Reconstruction Approach Based on Video Imagery. Frontiers in Marine Science, 2019, 6(96):1-21.

Point 2: Line 157: the accuracy of the laser scanner reported is incorrect: from the cited reference it is described as “10mm@50m”

Response 2: Corrected.

Point 3: Line 175: “completebess” > “completeness”

Response 3: Corrected.

Point 4: Figures 5 and 6: there is a reference to (a)-(e) but only sub-figures up to (d) are displayed

Response 4: Corrected.

Point 5: The References appear slightly dated. Apart from #18 from 2019, every other reference dates back to 2016. Considering that the subject is constantly evolving, some more recent innovations could be mentioned in the introduction.

Response 5: Corrected. The following publications in the field of change detection are referred in the Introduction section and added in the References section.

  1. Lin Y.; Zhang L.; Wang N.; et al. A change detection method using spatial-temporal-spectral information from Landsat images. International Journal of Remote Sensing, 2019, 41(2):1-22.
  2. Bolorinos J.; Ajami N. K.; Rajagopal R. Consumption Change Detection for Urban Planning: Monitoring and Segmenting Water Customers During Drought. Water Resources Research, 2020, 56(3).
  3. Matikainen L.; Pandi M.; Li F.; et al. Toward utilizing multitemporal multispectral airborne laser scanning, Sentinel-2, and mobile laser scanning in map updating. Journal of Applied Remote Sensing, 2019, 13(4).
  4. Wenzhuo L.; Kaimin S.; Deren L.; et al. A New Approach to Performing Bundle Adjustment for Time Series UAV Images 3D Building Change Detection. Remote Sensing, 2017, 9(6).
  5. Dong, P.L.; Zhong, R.F.; Yigit, A. Automated parcel-based building change detection using multitemporal airborne Lidar data. Surveying and Land Information Science, 2018, 77(1):5-13.
  6. Shirowzhan S.; Samad S.M.E.; Li H.; et al. Comparative analysis of machine learning and point-based algorithms for detecting 3D changes in buildings over time using bi-temporal lidar data. Automation in Construction, 2019, 105:102841.
  7. Santos, R.C.D.; Galo, M.; Carrilho, A.C.; et al. Automatic building change detection using multi-temporal airbone Lidar data. Proceedings of The International Archives of the Photogrammetry, Remote Sensing and Spatial Information Sciences, Volume XLII-3/W12-2020, 2020 IEEE Latin American GRSS & ISPRS Remote Sensing Conference, Santiago, Chile. 2020.

Point 6: In which development environment did you implement your method? Would it be interesting for you to release it to the scientific community on public domain platforms like GitHub?

Response 6: Thank you for your comments and suggestions. In this paper, this method was implemented on MATLAB 2016a. The platform of the experiments is a personal computer, equipped with a 2.50 GHz Intel Core i7-6500U CPU, 8 GB of main memory. We would like to release it to the scientific community on public domain platforms like GitHub. In order to facilitate the use of other academics, we are cleaning up the code so that it is a readable and extendable style.

Round 2

Reviewer 1 Report

The author improved the paper. Also, the value of findings, contributions, and the impact should be highlighted in the abstract.